# Cyclodextrin reduces cholesterol crystal uptake by circulating monocytes in patients undergoing coronary angiography

**Nikola Lübbering**[1]*, **Alexander Krogmann**[1], **Felix Jansen**[1], **Eicke Latz**[2], **Georg Nickenig**[1], **Sebastian Zimmer**[1]

1 Medizinische Klinik und Poliklinik II, Herzzentrum, Universitätsklinikum Bonn, Bonn, Germany,
2 Institute of Innate Immunity, Universitätsklinikum Bonn, Bonn, Germany

* nikola.luebbering@ukbonn.de

## Abstract

### Background

Atherosclerosis is a chronic inflammatory disease driven by endothelial dysfunction, cholesterol accumulation, and immune activation leading to thrombosis and vascular stenosis. While LDL-lowering therapies are firmly established, targeting the underlying inflammation is still an emerging strategy. Cholesterol crystals (CC) contribute to inflammation by activating the NLRP3 inflammasome in monocytes and promoting disease progression. Cyclodextrin (CD), an FDA-approved drug carrier, has shown atheroprotective effects by enhancing cholesterol metabolism and reducing inflammation in preclinical models. This study investigated whether CC-uptake in human monocytes, a prerequisite for inflammasome activation, is also influenced by CD pretreatment.

### Methods

Human peripheral mononuclear cells were isolated from whole blood samples provided by 76 patients undergoing coronary angiography at the University Hospital Bonn between November 2017 and February 2018. After separation, peripheral mononuclear cells were stimulated with 2-Hydroxypropyl-γ-Cyclodextrin and CC. CC-uptake by monocytes was analyzed using flow cytometry.

### Results

CC-uptake by monocytes varied greatly between patients (8–37%), with lower uptake observed in patients with elevated leukocytes ($p = 0.0058$) and diabetes mellitus ($p = 0.0448$). CD-pretreatment significantly reduced CC-uptake (20.1% ± 0.8% vs. 15.0% ± 0.6%, $p < 0.0001$). Interindividual variability in CD response (CCΔCD) was noted; 40 patients exhibited a significant reduction in CC-uptake, while nine showed

**Data availability statement:** All data files are available from the bonndata database (DOI: https://doi.org/10.60507/FK2/0SYROH).

**Funding:** This work was funded by the Deutsche Forschungsgemeinschaft (DFG, German Research Foundation) - grant no. 397484323 – TRR259 – A02 to G.N., S.Z. and E.L.

**Competing interests:** The authors have declared that no competing interests exist.

**Abbreviations:** ABCA1/-G1, ATP-binding Cassette Transporter-A1/G1; ACS, Acute Coronary Syndrome; AP, Angina Pectoris; AS, Aortic Valve Stenosis; ASA, Acetylsalicylic Acid; CAD, Coronary Artery Disease; CC, Cholesterol Crystals; CD, Cyclodextrin; CD11b, 14, 16, 45, Cluster Of Differentiation; CRP, C-reactive Protein; CVRF, Cardiovascular Risk Factors; EF, Ejection Fraction; FDA, U.S.-Food and Drug Administration; ICAM-1, Intercellular Adhesion Molecule 1; IL, Interleukin; LDL, Low-Density Lipoprotein; LXR, Liver-X-Receptor; NF-κB, Nuclear Factor Kappa-light-chain-enhancer Of Activated B-cells; NLRP, Nucleotide-binding Oligomerization domain, Leucine-rich Repeat and Pyrin Domain Containing; PBMC, Peripheral Mononuclear cells; PBS, Phosphat Buffered Saline; PCI, Percutaneous Coronary Intervention; VCAM-1, Vascular Cell Adhesion Molecule 1.

an increase. Patients with coronary artery disease (CAD) ($p = 0.0316$), requirement for percutaneous coronary intervention (PCI) ($p = 0.0030$), and elevated leucocyte levels ($p = 0.0135$) had lower CCΔCD, suggesting a link between systemic inflammation and attenuated CD efficacy.

## Conclusion

We demonstrated that CD significantly reduced CC-uptake in patients undergoing coronary angiography, which supports its role in inhibiting CC-phagocytosis and promoting cholesterol efflux. Interestingly, patient response to CD varied, with those exhibiting greater systemic inflammation or CAD showing a less pronounced reduction in CC-uptake. Our findings provide insight into the atheroprotective mechanisms of CD and suggest its potential utility in evaluating individual cardiovascular risk and monitoring CD-based therapeutic interventions in humans.

## Introduction

Atherosclerosis is a chronic inflammatory disease of the arterial wall. According to current disease models, classic cardiovascular risk factors damage the endothelium and impair its function. This promotes the accumulation of cholesterol in the sub-endothelial space, triggering chronic inflammation and leading to thrombosis and stenosis of arterial vessels, which can cause heart attacks, strokes, and peripheral vascular disease [1,2]. Collectively, these conditions represent the leading cause of death worldwide [3], underscoring the urgent need for novel therapeutic approaches to reduce disease risk. While current treatments primarily focus on reducing low-density lipoprotein (LDL) cholesterol, not all patients achieve adequate LDL reduction [4–6]. Recent studies are targeting the underlying chronic inflammation as a promising pharmaceutical strategy [7–9].

Excessive cholesterol accumulation in atherosclerotic lesions results in the formation of cholesterol crystals (CCs) deposited in the subendothelial extracellular space. The phagocytosis of CCs by invading monocytes causes lysosomal damage, which activates an inflammatory response via the NLRP3 inflammasome [10,11]. Thus, reducing CC-phagocytosis may help mitigate systemic inflammation and slow the progression of atherosclerotic lesions.

Human peripheral blood mononuclear cells (PBMCs), comprising lymphocytes and monocytes, are key indicators of inflammation. Both lymphocytes and monocytes play significant roles in atherosclerosis pathogenesis, with monocytes—particularly those that differentiate into macrophages—predominating in atherosclerotic lesions. Monocytes are a heterogeneous population, classified into three main subsets: classical (CD14++CD16−), intermediate (CD14++CD16+), and non-classical (CD14+CD16++). Classical and intermediate subsets have been shown to predict cardiovascular event rates, while Hamers et al. revealed heterogeneity within the non-classical subset. Certain subpopulations in this group are more frequent in severe coronary artery disease (CAD), suggesting potential therapeutic targets

[12–15]. In general, macrophages are programmed to contribute to the progression of advanced atherosclerotic lesions through excessive cholesterol uptake. Impaired cholesterol efflux further exacerbates cholesterol accumulation, creating a pro-inflammatory and pro-atherosclerotic environment [10,16–19].

Previous studies have demonstrated that cyclodextrin (CD), an FDA-approved compound used for delivering lipophilic drugs [20], has beneficial effects on atherogenesis. CD promotes macrophage reprogramming toward a more efficient cholesterol metabolism and an anti-inflammatory phenotype [21,22]. We now tested whether 2-Hydroxypropyl-γ-CD-pretreatment of human monocytes impacts CC-uptake and thereby affects the innate inflammatory stimulus.

## Materials and methods

### CC-preparation

For CC-preparation, 20 mg cholesterol was dissolved in 10 ml 1-propanol and subsequently incubated in 15 ml distilled water at room temperature for 10 min. A CC-pellet was produced by centrifugation at 3400 rpm for 10 min and dried in a vacuum centrifuge at 30 °C for 8–12 hours. Afterwards, CC were resuspended in 1 ml 0.1% fetal-bovine serum in phosphate-buffered saline (PBS) and finally treated in an ultrasonic bath for 10 min. Flow cytometry was used to determine the exact number of CC-particles.

### Sampling and stimulation

Whole blood samples were collected from 76 patients undergoing coronary angiography between 15/11/2017 and 14/02/2018. This study was approved by the University Bonn Ethics Committee on 25/06/2014. All participants provided written informed consent prior to enrolment in the study.

For separation of PBMC, a cell separation medium with a polyester gel and a density gradient liquid was used [23]. PBMC were dissolved in the patient`s own serum (7.2 ml) and diluted in (7.8 ml) PBS. The PBMC were preincubated with 2-Hydroxypropyl-γ-CD (10 mM) or PBS control for 6h at 37°C and subsequently incubated with CC (120 x 10^6 particles) for 30 min at 37°C. The concentration of 10mM CD was selected based on our own previously published data demonstrating adequate biological activity without cytotoxic effects at this dose [22]. CC-uptake by monocytes was evaluated and quantified by flow cytometry. To increase validity of the result, samples were stimulated and measured in triplets (S1 and S2 Figs). Cell count by flow cytometry resulted in approximately 100,000 cells per sample.

After stimulation with CD and CC, PBMC were treated with 12.5μl 4% paraformaldehyde in PBS. To differentiate monocytes from lymphocytes and cell debris, we used three different monoclonal antibodies: PE anti-human CD11b (Clone ICRF44, BioLegend, San Diego, CA, USA), FITC anti-human CD14 (Clone M5E2, BioLegend) und PerCP anti-human CD45 (Clone 2D1, BioLegend) (Dilution: 1 μg/test). We analyzed 10,000 cells per sample in a predefined PBMC-gate.

### Gating strategy

CD45 + Monocytes and lymphocytes were separated from CD45- cell debris. A "single-cell"-gate was used to remove duplets from further analysis [24]. Monocytes were separated from lymphocytes as CD14+/CD11b+-cells. CC-uptake was defined and quantified by a shift in side scatter, which reflects increased granularity after phagocytosis, as previously described by Samstad et al. [25]. For each patient, we acquired an unstimulated control sample (PBMC without CC-stimulation) to define the side scatter characteristics of monocytes in the absence of crystals. On this control, an individual, cell-free side scatter gate was placed directly above the population of monocytes. Monocytes incubated with CC showed a shift in side scatter into this predefined gate. The proportion of monocytes in this gate was further analyzed (Fig 1).

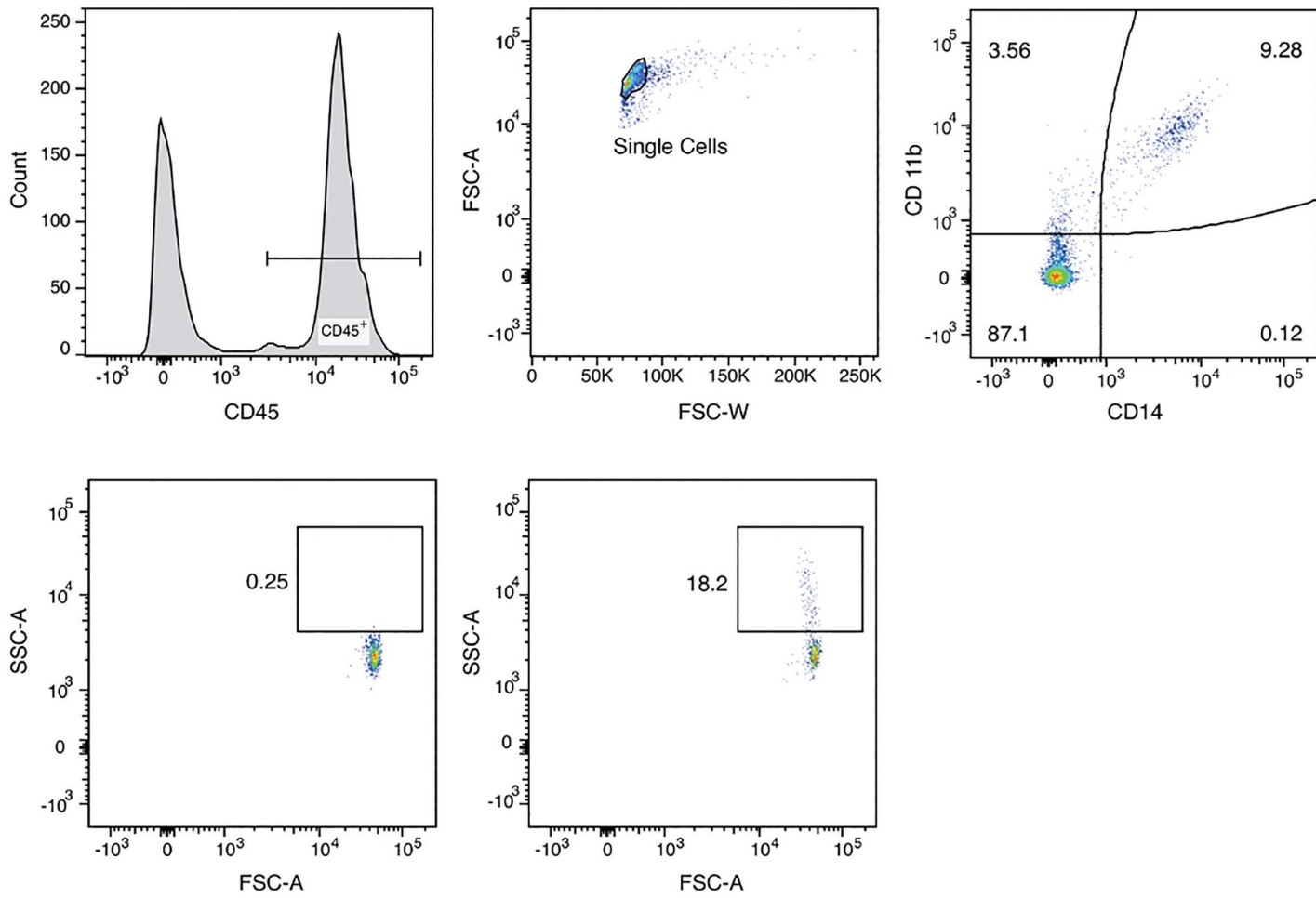

**Fig 1. Gating strategy.** 1.) CD45+ positive cells were selected. 2.) Single cells were identified and duplets excluded. 3.) CD14+/CD11b+ monocytes (upper right) were gated. 4.) A gate with higher SSC-A was selected using CD14+/CD11b+ monocytes without CC-incubation. 5.) SSC-A-Shift is observed upon CC-uptake by CD14+/CD11b+ monocytes. (CD 11b, 14, 45: Cluster of Differentiation, CC: Cholesterol Crystals).

## Statistics

Categorical variables were analyzed using Chi-squared tests. Post-hoc analyses were performed by Fisher`s exact test, and the data are presented as absolute numbers and percentages. Normality of continuous variables was assessed by Q-Q plots and the Shapiro-Wilk test. For normally distributed continuous variables, unpaired Student`s t-tests or ANOVA (for multiple comparisons) were used. Post-hoc comparisons were performed with the Sidak`s test. Data are presented as mean ± standard error of mean (s.e.m.). For non-normally distributed variables (Shapiro-Wilk $p < 0.05$), group comparisons were performed using the Mann-Whitney U test, and the data are presented as median [interquartile range, IQR]. $p < 0.05$ was considered statistically significant. Analyses were performed with Excel (Microsoft), Prism (GraphPad Software Inc.) and SPSS (IBM).

## Results

Whole blood samples were collected from 76 patients via the side port of an arterial sheath prior to coronary angiography. The average age of the patients included in our study was 71 years and 70% were male. Most were elective procedures

(72%). CAD was diagnosed in two thirds of patients (66%) and 29% required percutaneous coronary intervention (PCI). The mean left ventricular ejection fraction (EF) was 55.5%.

As previously described, CC-uptake results in an increase in granularity, which was defined as an increase in side scatter. As proof of concept, monocytes that were not previously incubated with CC did not show a change in side scatter (p<0.0001) (Fig 2A). The relative number of monocytes that incorporated CC ranged between 8 and 37%. To evaluate whether specific patients` characteristics were associated with these interindividual differences, we divided the collective into two groups based on the median of the gaussian distribution of the CC-uptake (20%) (Fig 2B). Patients with elevated plasma levels of leukocytes (p=0.0058) or diabetes mellitus (p=0.0448) had a significant lower CC-uptake (Table 1 and S1 Table).

We next examined if the CC-uptake of monocytes is affected by pretreatment with CD. The CCΔCD was defined as the difference between CC-uptake of monocytes incubated with PBS control and CC-uptake of monocytes that were previously stimulated with CD. A higher CCΔCD consequently shows a lower CC-uptake after CD-incubation. CD-pretreatment significantly reduced CC-uptake overall (PBS+40 µl CC in PBS: 20.1%±0.8% vs. 10mM CD+40 µl CC in PBS: 15.0%±0.6%, p<0.0001). As proof of concept, PBMC that were not incubated with CCs did not show an increase in granularity after CD-stimulation (Fig 3A).

The CCΔCD showed interindividual variance. PBMC derived from 40 patients demonstrated an individually significant CCΔCD, whereas cells from nine patients exhibited an increase in CC-uptake following CD-stimulation. To understand these interindividual differences in PBMC phenotype, we compared patients` characteristics according to their CCΔCD. Patients were divided into two groups based on the median of the gaussian distribution of the CCΔCD (5.6%) (Fig 3B). Significantly more patients with CAD (p=0.0316), PCI (p=0.0030) or elevated plasma levels of leucocytes (p=0.0135) showed an attenuated CCΔCD (< 5.6%), indicating that the CCΔCD could potentially function as an indicator for higher inflammation, CAD or the requirement for PCI (Table 2). Within the subgroup of patients with CAD, individuals with three-vessel disease were significantly more frequent in the attenuated CCΔCD group (p=0.0216), whereas one- and two-vessel disease were not overrepresented, suggesting that a higher plaque burden may be associated with a reduced CCΔCD (Table 2 and S2 Table).

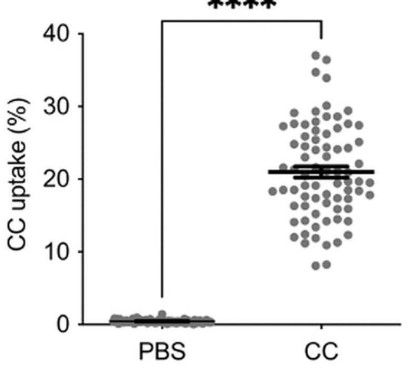
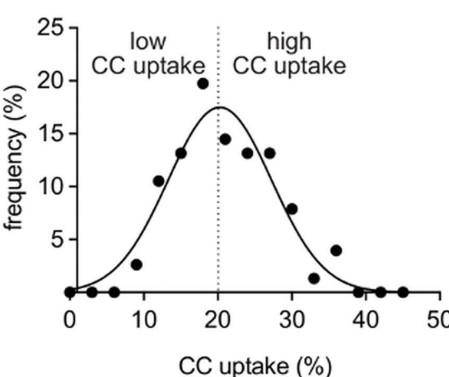

**Fig 2. CC-uptake after CC-incubation. A)** Monocytes showed a significantly higher CC-uptake after CC-incubation than after incubation with PBS-control (p<0.0001). **B)** Gaussian distribution of CC-uptake to divide the total collective into two groups (low and high CC-uptake) according to the median at 20%. (CC: Cholesterol Crystals, PBS: Phosphate-buffered saline).

**Table 1. Characteristics of patients in relation to CC-uptake.**

| | Total collective | CC-uptake > 20% | CC-uptake < 20% | p-value |
|---|---|---|---|---|
| | N = 76 | N = 38 | N = 38 | |
| Age | | | | |
| Mean – yr | 71.0 ± 10.6 | 72.1 ± 10.5 | 69.6 ± 10.9 | 0.3126 |
| ≥ 65 yr – no. (%) | 54 (71) | 28 (74) | 26 (68) | 0.4004 |
| Male sex – no. (%) | 53 (70) | 27 (71) | 26 (68) | 0.5000 |
| Structural Parameters | | | | |
| EF (%) – median [IQR] | 59.5 [50.0–65.0] | 55.0 [49.0–65.0] | 60.0 [55.0–62.0] | 0.5171 |
| AS | 16 (21) | 9 (24) | 7 (18) | 0.3446 |
| Reason for Coronary Angiography – no. (%) | | | | |
| ACS | 14 (18) | 5 (13) | 9 (24) | 0.1877 |
| Stable AP | 7 (9) | 4 (11) | 3 (8) | 0.5000 |
| Elective | 55 (72) | 29 (76) | 26 (68) | 0.3043 |
| Coronary Parameters – no. (%) | | | | |
| CAD | 50 (66) | 27 (71) | 23 (61) | 0.2343 |
| 1-vessel | 9 (12) | 5 (13) | 4 (11) | 0.5000 |
| 2-vessel | 11 (14) | 5 (15) | 6 (16) | 0.5000 |
| 3-vessel | 30 (39) | 17 (45) | 13 (34) | 0.2409 |
| PCI | 29 (28) | 14 (37) | 15 (39) | 0.5000 |
| CVRF – no. (%) | | | | |
| Hypertension | 67 (88) | 36 (95) | 31 (82) | 0.0763 |
| Diabetes mellitus | 26 (34) | 9 (24) | 17 (45) | **0.0448** |
| Dyslipidemia | 50 (66) | 28 (74) | 22 (58) | 0.1132 |
| Active Smoker | 23 (30) | 10 (26) | 13 (34) | 0.3090 |
| Lab Results – median [IQR] | | | | |
| LDL (mg/dl) | 103.0 [80.0–130.8] | 84.0 [79.5–112.0] | 118.0 [82.0–152.0] | 0.0503 |
| CRP (mg/dl) | 2.3 [1.0–10.6] | 1.8 [0.7–7.7] | 3.4 [1.1–13.0] | 0.1134 |
| Leucocytes (G/l) | 7.4 [5.8–8.6] | 6.0 [5.2–7.9] | 7.9 [6.5–10.7] | **0.0058** |

Patients with lower CC-uptake < 20% show significantly higher leucocyte counts (p = 0.0058) and more often have diabetes mellitus (p = 0.0448). (ACS: Acute Coronary Syndrome, AP: Angina pectoris, AS: Aortic Valve Stenosis, CAD: Coronary Artery Disease, CC: Cholesterol Crystals, CRP: C-reactive Protein, CVRF: Cardiovascular Risk Factors, EF: Ejection Fraction, LDL: Low-Density Lipoprotein, PCI: Percutaneous Coronary Intervention)

To further analyze the CCΔCD, we compared the collective according to the requirement for PCI. Patients requiring PCI showed a significantly attenuated CCΔCD (p = 0.0004) (Fig 4).

## Discussion

In this study, we tested the hypothesis that CD affects CC-phagocytosis by circulating monocytes. Pretreatment of human monocytes with CD significantly reduced CC-uptake in patients undergoing coronary angiography. A potential mechanism underlying this reduction is decreased complement activation and reduced expression of complement receptors on the monocyte surface, leading to diminished CC-phagocytosis [21]. Additionally, our own previously published work has shown that CD increases CC-solubility and promotes the formation of oxysterols, which activate liver-X-receptor (LXR) signaling and induce an atheroprotective transcriptional program in macrophages [22].

Atheroprotective effects of CD have been shown to be LXR-dependent, as CD upregulates LXR target genes, such as ABCA1 and ABCG1, to stimulate reverse cholesterol transport. Furthermore, cholesterol efflux has been observed

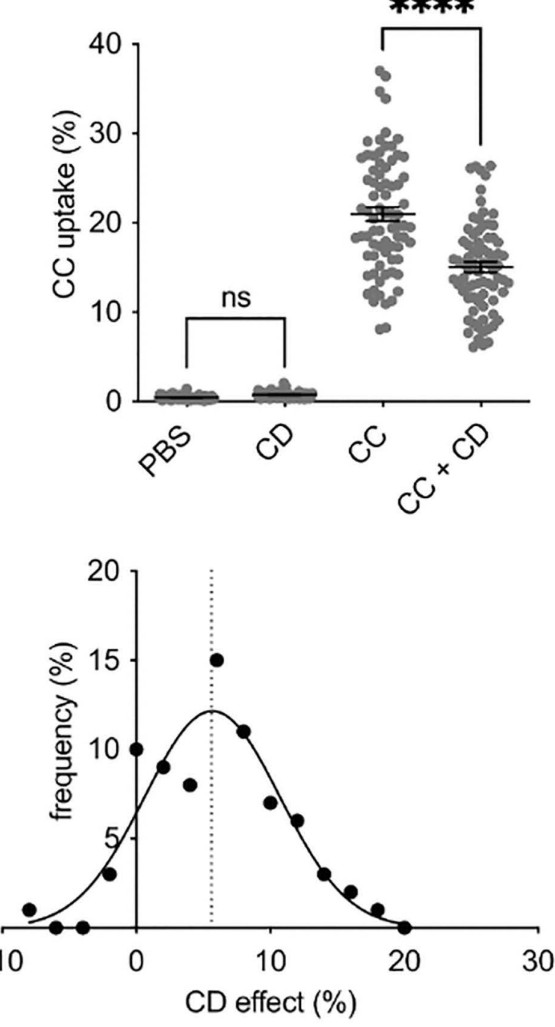

**Fig 3. CC-uptake after CD-stimulation. A)** CD-pretreatment significantly reduced CC-uptake overall (PBS + 40 µl CC in PBS: 20.1% ± 0.8% vs. 10mM CD + 40 µl CC in PBS: 15.0% ± 0.6%, P < 0.0001). PBMC that were not incubated with CC did not show an increase in granularity after CD-stimulation. **B)** Gaussian distribution of CCΔCD to divide the total collective into two groups (low and high CCΔCD) according to the median at 5.6%. (CC: Cholesterol Crystals, CD: Cyclodextrin, PBMC: Peripheral Mononuclear Cells, PBS: Phosphat-buffered Saline).

even in ABCA1- and ABCG1-deficient mice, suggesting the involvement of alternative pathways to reduce CC-burden in monocytes and macrophages [22,26–29]. Beyond metabolic effects, LXRs exhibit anti-inflammatory properties, as their activation suppresses NF-κB and NLRP3, reducing IL-1β production [17,22,26,28,30–32]. As CC-phagocytosis activates the NLRP3 inflammasome and triggers IL-1β release, reducing CC-uptake in circulating monocytes may have significant anti-inflammatory potential [11,33].

Although CD reduced CC-uptake overall, the magnitude of reduction (CCΔCD) varied considerably among patients. We explored why some patients exhibited a pronounced reduction in CC-uptake after CD stimulation, while others showed little to no effect. Analysis of medical characteristics revealed that patients with negligible CCΔCD more frequently had CAD, required coronary intervention, and displayed higher plasma leukocyte levels. These findings suggest that patients with greater systemic inflammation, indicated by elevated leukocytes, or with a higher CAD burden, respond differently to CD

**Table 2. Characteristics of patients in relation to CCΔCD.**

| | Total collective | CCΔCD > 5.6% | CCΔCD < 5.6% | p-value |
|---|---|---|---|---|
| | N = 76 | N = 40 | N = 36 | |
| Age | | | | |
| Mean – yr | 71.0 ± 10.6 | 70.9 ± 10.5 | 70.1 ± 11.2 | 0.9342 |
| ≥ 65 yr – no. (%) | 54 (71) | 27 (68) | 27 (75) | 0.3212 |
| Male sex – no. (%) | 53 (70) | 29 (73) | 24 (67) | 0.3807 |
| Structural Parameters | | | | |
| EF (%) – median [IQR] | 59.5 [50.0–65.0] | 60.0 [50.8–65.0] | 56.5 [42.5–60.0] | 0.1556 |
| AS | 16 (21) | 12 (26) | 4 (13) | 0.1478 |
| Reason for Coronary Angiography – no. (%) | | | | |
| ACS | 14 (18) | 7 (18) | 7 (19) | 0.5296 |
| Stable AP | 7 (9) | 3 (8) | 4 (11) | 0.4402 |
| Elective | 55 (72) | 30 (75) | 25 (69) | 0.3878 |
| Coronary Parameters – no. (%) | | | | |
| CAD | 50 (66) | 22 (55) | 28 (78) | **0.0316** |
| 1-vessel | 9 (12) | 6 (15) | 3 (8) | 0.2962 |
| 2-vessel | 11 (14) | 5 (13) | 6 (17) | 0.4239 |
| 3-vessel | 30 (39) | 11 (28) | 19 (53) | **0.0216** |
| PCI | 29 (28) | 9 (23) | 20 (56) | **0.0030** |
| CVRF – no. (%) | | | | |
| Hypertension | 67 (88) | 37 (93) | 30 (83) | 0.1899 |
| Diabetes mellitus | 26 (34) | 13 (33) | 13 (36) | 0.4640 |
| Dyslipidemia | 50 (66) | 24 (60) | 26 (72) | 0.1898 |
| Smoker | 23 (30) | 11 (28) | 12 (33) | 0.3807 |
| Lab Results – median [IQR] | | | | |
| LDL (mg/dl) | 103.0 [80.0–130.8] | 107.0 [80.0–137.3] | 93.0 [79.3–121.0] | 0.3052 |
| CRP (mg/dl) | 2.3 [1.0–10.6] | 1.8 [1.1–9.9] | 3.6 [1.0–12.8] | 0.4814 |
| Leucocytes (G/l) | 7.4 [5.8–8.6] | 6.4 [5.3–8.2] | 7.8 [6.1–10.6] | **0.0135** |

Patients without a significant CCΔCD have significantly more often CAD (p = 0.0316), need PCI (p = 0.0030) and show high plasma levels of leucocytes (p = 0.0135). (ACS: Acute Coronary Syndrome, AP: Angina pectoris, AS: Aortic Valve Stenosis, CAD: Coronary Artery Disease, CC: Cholesterol Crystals, CRP: C-reactive Protein, CVRF: Cardiovascular Risk Factors, EF: Ejection Fraction, LDL: Low Density Lipoprotein, PCI: Percutaneous Coronary Intervention)

treatment ex vivo. While both acute and chronic inflammation are known to activate the complement system, it remains unclear whether complement suppression by CD directly contributes to the observed reduction in CC-uptake.

Previous studies have indicated that chemical modifications of CD influence its ability to reduce CC-uptake by monocytes. For instance, α-CD has shown similar effects, but 2-Hydroxypropyl-β-CD failed to reduce CC-uptake in a separate trial [21,34]. Notably, CD pretreatment duration varied between studies, with longer incubation times yielding stronger effects. In the present study, PBMC were preincubated with CD for six hours before CC-stimulation to allow sufficient interaction with monocytes. On the basis of our group`s previously published work on CD-induced LXR activation, it is plausible that prolonged exposure facilitates intracellular action of CD and could enable LXR-dependent gene regulation; however, intracellular CD-uptake and transcriptional responses were not directly assessed here and therefore remain speculative [22].

Interestingly, baseline CC-uptake by monocytes also varied among patients. Those with lower CC-uptake exhibited significantly higher plasma leukocyte levels and were more likely to have diabetes mellitus, suggesting that subliminal

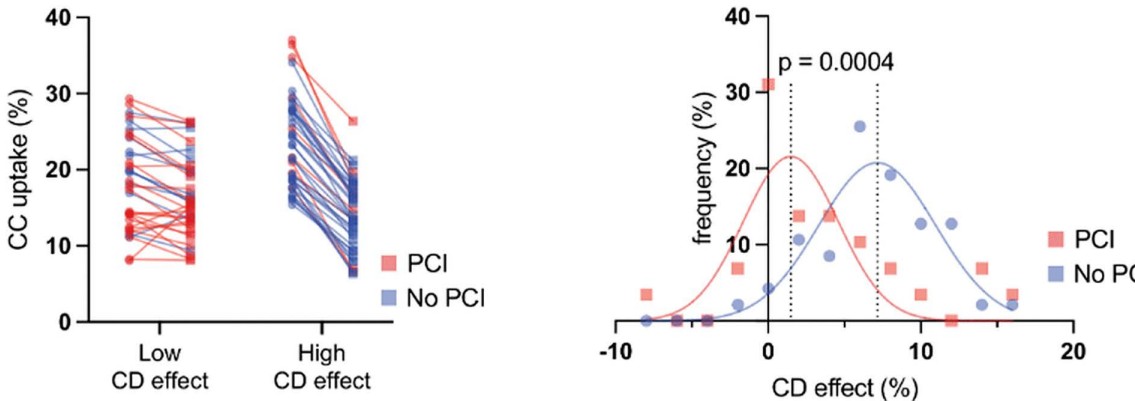

**Fig 4. CCΔCD and the requirement for PCI. A)** Patients are divided into two groups – low and high CCΔCD. In each group, CC-uptake before and after CD-stimulation is visualized. Numbers of patients requiring PCI are significantly higher in the CCΔCD-group. **B)** CCΔCD showed great interindividual variance. Patients who required PCI showed a significantly attenuated CCΔCD (p = 0.0004). (CC: Cholesterol Crystals, CD: Cyclodextrin, PCI: Percutaneous Coronary Intervention).

systemic inflammation may reduce CC-uptake. This finding contrasts with data from Pilely et al., who demonstrated increased CC-phagocytosis in the presence of acute-phase proteins and complement activation [35]. To assess whether the observed differences in CC-uptake could be explained by treatment rather than by patient characteristics, we analyzed concomitant cardiovascular and metabolic medications (S3 and S4 Tables). The use of antiplatelet therapy, oral anticoagulation, statins and guideline-directed antihypertensive therapy was comparable between patients with low and high CC-uptake, arguing against a major medication-related confounder of the baseline CC-uptake.

CD may represent a potential candidate for further investigation as a therapeutic approach in atherosclerosis, with documented benefits including reduced atherosclerotic lesion size, decreased CC-burden, plaque regression, enhanced reverse cholesterol transport, and reduced systemic inflammation [21,22].

While reducing CC-phagocytosis represents one aspect of CD's potential, a comprehensive understanding of its anti-inflammatory properties necessitates further investigation. Specifically, genetic analysis of monocytes following CD stimulation is essential to assess potential LXR induction, increased expression of abca1 and -g1, and decreased nlrp3 expression, which could result in reduced cytokine secretion.

## Limitations

This study has several limitations. First, the time between blood collection and sample processing ranged from 15 minutes to 3 hours, potentially affecting cell stability. Second, plasma monocyte levels varied between patients, yet a fixed number of PBMCs was used for each experiment. Third, only a single CD concentration (10 mM) was tested; it remains unclear whether different concentrations might yield distinct effects. Furthermore, we did not include a dedicated live/dead staining in the flow-cytometric assay; although 10 mM CD has been shown to be subtoxic in our previous work [22], an influence of non-viable cells on side-scatter-based quantification of CC-uptake cannot be fully excluded. In addition, information on concomitant cardiovascular and metabolic medication was obtained from clinical charts and not dose-standardized, so minor medication-related confounding cannot be fully excluded. In 71 of 76 patients, blood collection was performed before the administration of ASA (Acetylsalicylic acid) and/or heparin. The remaining 5 patients presented with non-ST-elevation myocardial infarction and received ASA and heparin prior to blood collection. An ASA/Heparin-related confounder therefore cannot be completely ruled out; however, given the small number of affected cased, a relevant bias appears unlikely.

Our ex vivo assay specifically assessed phagocytosis of cholesterol crystals by circulating monocytes. We did not investigate uptake or signaling in response to vascular calcium deposits. This is a limitation with regard to the full spectrum of coronary atherosclerotic plaque morphology. However, experimental work has shown that crystalline cholesterol is a potent trigger of monocyte and macrophage activation and NLRP3 inflammasome signaling, whereas monocytes do not phagocytose macro calcified plaque components to a comparable extent [11]. Our focus in this study was therefore on cholesterol crystals as the biologically relevant particulate stimulus for circulating monocytes.

## Conclusion

Our study examined the effect of CD on CC-phagocytosis by monocytes, revealing CD significantly reduces CC-uptake. Interestingly, patient responses to CD varied, with those exhibiting greater systemic inflammation or coronary artery disease showing a less pronounced reduction in CC-uptake. Our finding provide insight into the atheroprotective mechanisms of CD and suggest its potential utility in evaluating individual cardiovascular risk and monitoring CD-based therapeutic interventions in humans.

## Supporting information

**S1 Fig. Experimental Setup.** PBMC were stimulated with CD and incubated at 37°C for 6 hours. Afterwards they were stimulated with CC at 37°C for another 30 min. CC-uptake was analysed by Flow cytometry. (CC: Cholesterol Crystals, CD: Cyclodextrin, PBMC: Peripheral Mononuclear Cells).
(PDF)

**S2 Fig. Stimulation.** 12 samples per patient – 6 stimulated with CD and 6 incubated with PBS-control. In each group triplets stimulated with CC and 3 samples incubated with PBS-control. (CD: Cyclodextrin, PBMC: Peripheral Mononuclear Cells, PBS: Phophat Buffered Saline).
(PDF)

**S1 Table. Cholesterol levels in patients in relation to CC-uptake.** Patients with higher CC-uptake tended to have lower LDL, although this did not reach statistical significance. No differences in HDL or total cholesterol were observed in relation to CC-uptake. Data are presented as median (IQR) and groups were compared using the Mann–Whitney U test. (CC: Cholesterol Crystals, LDL: Low-Density Lipoprotein, HDL: High-Density Lipoprotein).
(PDF)

**S2 Table. Cholesterol levels in patients in relation to CCΔCD.** No differences in LDL, HDL or total cholesterol were observed across CCΔCD groups. Data are presented as median (IQR) and groups were compared using the Mann–Whitney U test. (CC: Cholesterol Crystals, LDL: Low-Density Lipoprotein, HDL: High-Density Lipoprotein).
(PDF)

**S3 Table. Concomitant medication according to CC-uptake.** No significant differences in medication were observed between patients with low and high CC-uptake (all p > 0.05). Data are shown as n (%). (CC: Cholesterol Crystals, ASA: Acetylsalicylic Acid, DAPT: Dual Antiplatelet Therapy, VKA: Vitamin K Antagonist, ARB: Angiotensin receptor blocker, PPI: Proton Pump Inhibitor).
(PDF)

**S4 Table. Concomitant medication according to CCΔCD.** Medication profiles were comparable across CCΔCD (all p > 0.05). Data are shown as n (%). (ASA: Acetylsalicylic Acid, DAPT: Dual Antiplatelet Therapy, VKA: Vitamin K Antagonist, ARB: Angiotensin receptor blocker, PPI: Proton Pump Inhibitor).
(PDF)

## Acknowledgments

We appreciate the great technical assistance of S. Adler and K. Groll (University of Bonn). 2-Hydroxypropyl-γ-Cyclodextrin was generously supplied by Wacker Chemie AG, Pharma Division.

## Author contributions

**Conceptualization:** Alexander Krogmann, Sebastian Zimmer.

**Data curation:** Nikola Lübbering, Sebastian Zimmer.

**Formal analysis:** Nikola Lübbering, Sebastian Zimmer.

**Funding acquisition:** Felix Jansen, Eicke Latz, Sebastian Zimmer.

**Investigation:** Nikola Lübbering.

**Methodology:** Nikola Lübbering, Alexander Krogmann, Felix Jansen, Sebastian Zimmer.

**Project administration:** Georg Nickenig, Sebastian Zimmer.

**Resources:** Georg Nickenig.

**Supervision:** Eicke Latz, Georg Nickenig.

**Visualization:** Sebastian Zimmer.

**Writing – original draft:** Nikola Lübbering, Sebastian Zimmer.

**Writing – review & editing:** Georg Nickenig.

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
