## [Decision Letter · Decision Letter 0]

29 Oct 2025

PONE-D-25-33939Cyclodextrin reduces cholesterol crystal uptake by circulating monocytes in patients undergoing coronary angiography.PLOS ONE

Dear Dr. Luebbering,

Thank you for submitting your manuscript to PLOS ONE. After careful consideration, we feel that it has merit but does not fully meet PLOS ONE’s publication criteria as it currently stands. Therefore, we invite you to submit a revised version of the manuscript that addresses the points raised during the review process. Please be sure to address all the points raised by the expert reviewers and quoted below. 

We look forward to receiving your revised manuscript.

Kind regards,

Marc W. Merx, MD

Academic Editor

PLOS ONE

Journal Requirements:

Reviewers' comments:

Reviewer's Responses to Questions

**Comments to the Author**

1. Is the manuscript technically sound, and do the data support the conclusions?

Reviewer #1: Yes

Reviewer #2: Partly

2. Has the statistical analysis been performed appropriately and rigorously?

Reviewer #1: Yes

Reviewer #2: No

3. Have the authors made all data underlying the findings in their manuscript fully available?

Reviewer #1: Yes

Reviewer #2: Yes

4. Is the manuscript presented in an intelligible fashion and written in standard English?

Reviewer #1: Yes

Reviewer #2: Yes

5. Review Comments to the Author

Reviewer #1: The authors investigated uptake of cholesterol crystals of human monocytes of patients undergoing coronary angiography and the influence of cyclodextrin treatment. They observed reduction of cholesterol crystal uptake under cyclodextrin treatment with different response depending on individual patients risk factors providing import evidence for cyclodextrin as potential therapeutic strategy in the future to reduce CV events. I recommend to accept the manuscript with major revisions stated below:

1. The authours should provide information on concomitant medication of investigated patients and should discuss potential medication confounders affecting uptake of cholesterol crystals. They also should discuss potential influence of medication used during acute coronary syndrome like aspirin or heaprine.

2. Was a dead-and-alive staining included into the flow cytometry analyisis? If not please comment on this in the discussion section as dead cells included in the analysis might influence SSC shift for cholesterol crystal uptake.

Reviewer #2: The authors present an ex-vivo analysis of human PBMC investigating cyclodextrin, an FDA-approved drug carrier, has shown atheroprotective effects by enhancing cholesterol metabolism and reducing inflammation and cholesterol crystals. The authors show that cyclodextrin reduces cholesterol crystals uptake in patients undergoing coronary angiography. They hypothesize that these findings supports the role of cyclodextrin in inhibiting phagocytosis of cholesterol crystals and thus, promoting cholesterol efflux. Further, the authors state that observed ex-vivo effects are due to transcriptional changes

While the topic is highly relevant, there are major flaws that limit the quality of the manuscript.

Major remarks

1. The authors use circulating monocytes for the experiments. This model is significantly flawed, as resident cells in plaque tissue may have different characteristics. Circulating monocytes are usually not in contact with cholesterol crystals, thus greatly limiting the biological rigor of the finding.

2. Sideward scatter is a very general measure that can indicate a lot of things, e.g., different granules. The authors do not show that the increase in sideward scatter is in fact induced by cholesterol. This must be demonstrated.

3. The authors should provide data on the association of CC uptake and plaque burden and cholesterol levels in patients.

4. Calcified plaques are the most common finding in coronary atherosclerosis – yet, calcium was not assessed here which is a significant limitation towards the real situation.

5. The statistical analysis lacks assessment of normal distribution and appropriate testing of non-normally distributed values

6. Page 8: The sentence “CD-143 incubation of PBMC for six hours before CC-stimulation allows for internalization of CD by monocytes and may lead to transcriptional changes as described above” should also be moved to the Discussion section. Since internalization of CD was not directly validated (only surface markers were assessed via FACS), this causal claim cannot be supported by the current data. Likewise, any mention of transcriptional changes should be either substantiated with transcriptomic evidence or clearly stated as a hypothesis based on previously published findings (with appropriate references).

7. Calculation of CCΔCD introduces another step of calculation that may lead to bias. The authors should argue, based on current literature, why this value was introduced rather than showing normalized or raw data.

8. In several points in the manuscript, the authors refer to gene expression changes, this is not backed up by the data and should be removed.

9. The conclusion “CD has shown potential as a novel therapeutic strategy for atherosclerosis” should be reframed. Given that the current study is based on ex vivo analyses, such a therapeutic claim is not substantiated. A more appropriate phrasing would acknowledge that “CD may represent a potential candidate for further investigation as a therapeutic approach in atherosclerosis.”

Minor remarks

1. Please include standard deviations and percentual values where appropriate throughout the manuscript. Additionally, indicate the sample size (n = xy) adjacent to percentage values for transparency.

2. Please rephrase the sentence “The relative number of monocytes that incorporated CC varied individually between 8 and 37%” to “The relative number of monocytes that incorporated CC ranged between 8–37%.”

3. Page 8: The statement “which validates the hypothesis that CD reduces CC-phagocytosis” should be relocated to the Discussion section, as it represents an interpretation rather than a result.

4. The statement “40 patients displayed an individually significant CCΔCD while nine patients developed an increase in CC-uptake after CD-stimulation” requires revision. As the experiments were conducted ex vivo, it would be more accurate to state that “PBMCs derived from 40 patients demonstrated a significant CCΔCD, whereas cells from nine patients exhibited an increase in CC uptake following CD stimulation.”

5. Throughout the manuscript, please replace the term “heart catheterization” with the more precise and clinically appropriate term “coronary angiography.”

6. The sentence “Additionally, CD increases CC solubility and promotes the formation of oxysterols, which activate liver-X-receptor (LXR)” should be either supported by a reference or removed, as there are no LXR-related experiments reported in the current study.

6. PLOS authors have the option to publish the peer review history of their article (what does this mean?). If published, this will include your full peer review and any attached files.

Reviewer #1: No

Reviewer #2: No

---

## [Author Response · Author response to Decision Letter 1]

12 Nov 2025

Dear Reviewers,

Thank you for the thorough and constructive assessment of our manuscript. We have revised the text accordingly and respond to each comment point by point below. Where clarifications were requested, we have added them to the Methods, Results, Discussion or Supplementary material.

Reviewer 1

1. The authours should provide information on concomitant medication of investigated patients and should discuss potential medication confounders affecting uptake of cholesterol crystals. They also should discuss potential influence of medication used during acute coronary syndrome like aspirin or heaprine.

We appreciate this comment, because medication is indeed a potential source of confounding in assays that assess monocyte function ex vivo. We therefore reviewed the medication records of all 76 patients and grouped drugs into clinically meaningful categories (antiplatelet therapy, oral anticoagulation, ACE inhibitor/ARB, β-blocker, calcium-channel blocker, diuretics, statin, antidiabetic therapy, steroids, proton pump inhibitor). We then examined CC uptake and the CC�CD across these medication groups. The results did not show any significant differences across groups and are now provided in Supplementary Tables S3 and S4. We have added a short paragraph to the Discussion to state explicitly that concomitant cardiovascular and metabolic medication was comparable between patients with low and high CC uptake.

With regard to acute antithrombotic therapy, we would like to clarify the timing. No STEMI patients were included. Five patients presented with NSTEMI and received aspirin plus intravenous heparin before coronary angiography and before study blood sampling. All remaining 71 of 76 patients were elective cases, and in these patients blood for the study was drawn before periprocedural aspirin and/or heparin were administered. Thus, while we cannot completely rule out an aspirin/heparin-related confounder in the small NSTEMI subset, a systematic medication-induced bias for the overall cohort appears unlikely. We have added this clarification to the Study Limitations.

2. Was a dead-and-alive staining included into the flow cytometry analysis? If not please comment on this in the discussion section as dead cells included in the analysis might influence SSC shift for cholesterol crystal uptake.

We appreciate this thoughtful comment. We did not include a separate live/dead staining in the original panel. Our choice of 10 mM cyclodextrin was based on our group’s previously published work (Zimmer et al., 2016), where this CD-concentration was biologically effective but did not show relevant toxicity in macrophages. We have now added a sentence to the Methods to clarify that 10 mM CD was selected because it had been shown to be subtoxic in our previous experiments. In addition, we listed the missing viability staining as a limitation.

Reviewer 2

Major remarks

1. The authors use circulating monocytes for the experiments. This model is significantly flawed, as resident cells in plaque tissue may have different characteristics. Circulating monocytes are usually not in contact with cholesterol crystals, thus greatly limiting the biological rigor of the finding.

We agree that plaque-resident macrophages and circulating monocytes are not identical and that the plaque microenvironment further shapes macrophage function. In the present study, however, our intention was to interrogate circulating monocytes because several cardiovascular conditions are known to systematically “prime” blood monocytes toward a more proinflammatory or cholesterol-sensitive state (e.g., in coronary artery disease, diabetes or acute coronary syndromes, circulating monocytes show altered subset composition, higher expression of activation markers and enhanced inflammasome responsiveness). In other words, disease-related conditioning already occurs in the blood compartment and can be captured ex vivo. Using PBMC from patients therefore allowed us to study these interindividual, disease-related differences upstream of plaque infiltration and to test whether CD can still reduce CC-induced activation under such conditions. We fully acknowledge that this approach does not reproduce the full complexity of plaque-resident macrophages, but it was chosen deliberately to focus on circulating monocytes as the clinically accessible compartment.

2. Sideward scatter is a very general measure that can indicate a lot of things, e.g., different granules. The authors do not show that the increase in sideward scatter is in fact induced by cholesterol. This must be demonstrated.

We apologize for the confusion – we realize that our original description was too brief. We have therefore expanded the gating strategy in “Methods”. As in Samstad et al., 2014, we first acquired a CC-free control from the same patient and placed an individual sideward scatter-gate directly above the monocyte population on this control, i.e. at a level where monocytes without crystals do not appear. After incubation with CC, only monocytes that had taken up CC shifted into this predefined gate, so the sideward scatter-increase reflects CC-dependent phagocytosis. Figure 1 illustrated this gating strategy, and Figure 2A shows monocytes without cholesterol crystal stimulation did not exhibit a shift in sideward scatter.

3. The authors should provide data on the association of CC uptake and plaque burden and cholesterol levels in patients.

We appreciate this comment and agree that linking CC-uptake to the extent of coronary artery disease and to circulating lipid levels strengthens the clinical relevance of the assay. In this cohort we used the number of diseased coronary vessels (no CAD, 1-, 2-, 3-vessel CAD) as a pragmatic proxy for plaque burden. CC-uptake and CC�CD were already tabulated across these angiographic categories in the original manuscript (Tables 1 and 2). To make this association more transparent, we have now added a sentence to the Results stating that, among patients with CAD, those with three-vessel disease were significantly more often in the group with an attenuated CC�CD, whereas one- and two-vessel disease were not overrepresented. No association was observed between plaque burden and baseline CC-uptake.

In addition, beyond the LDL values that were already reported in the manuscript, we also extracted HDL and total cholesterol and compared these lipid parameters between patients with low versus high CC-uptake and low versus high CC�CD. These results are now provided in Supplementary Tables S1 and S2. In all comparisons, we did not observe significant differences in LDL, HDL, or total cholesterol between the respective CC-uptake or CC�CD groups.

4. Calcified plaques are the most common finding in coronary atherosclerosis – yet, calcium was not assessed here which is a significant limitation towards the real situation.

We agree that human coronary atherosclerosis frequently contains calcified components. In the present study we deliberately focused on CC, because circulating monocytes are able to phagocytose CC and mount an NLRP3-dependent inflammatory response, as shown by Duewell et al. 2010., while macrocalcified plaque material is not phagocytosed by monocytes to the same extent. Our assay was designed to model this CC-driven interaction and not the full histological heterogeneity of coronary plaques. We have now added a sentence to the Limitations to clarify that only CC, but not calcium deposits, were assessed in this work.

5. The statistical analysis lacks assessment of normal distribution and appropriate testing of non-normally distributed values.

You are right – in the original version we did not explicitly test all continuous variables for normality. We have now repeated the analyses with assessment of normal distribution using Q–Q plots and the Shapiro–Wilk test. Non-normally distributed variables (Ejection fraction, leukocytes, CRP, LDL, HDL, total cholesterol) were re-analyzed using the Mann–Whitney U test, and are now reported as median [IQR]. The corresponding parts in the Methods and Results sections were updated accordingly; these changes did not significantly alter the main conclusion of the study.

6. Page 8: The sentence “CD-143 incubation of PBMC for six hours before CC-stimulation allows for internalization of CD by monocytes and may lead to transcriptional changes as described above” should also be moved to the Discussion section. Since internalization of CD was not directly validated (only surface markers were assessed via FACS), this causal claim cannot be supported by the current data. Likewise, any mention of transcriptional changes should be either substantiated with transcriptomic evidence or clearly stated as a hypothesis based on previously published findings (with appropriate references).

Thank you for pointing this out. We have removed the sentence from the Results section and placed a reworded version in the Discussion, where it is treated as an interpretation rather than as a finding. In the revised text we now state that the 6h-preincubation with 10mM CD was chosen to allow sufficient interaction with monocytes and that, based on our group`s previously published work (Zimmer et al., 2016), an LXR-dependent mechanism is plausible. We also explicitly clarify that in the present study we did not directly demonstrate intracellular CD uptake or CD-induced gene expression, and that these mechanisms should therefore be regarded as hypothetical.

7. Calculation of CCΔCD introduces another step of calculation that may lead to bias. The authors should argue, based on current literature, why this value was introduced rather than showing normalized or raw data.

Thank you for highlighting this. Our intention with CCΔCD was to capture the within-patient effect of CD despite the very large interindividual variability in baseline CC-uptake. Baseline uptake differed markedly between patients, so expressing the effect of CD as a delta relative to each patient`s own control allows us to compare patients on a common scale and to show how strongly CD modulates CC-uptake in that specific individual. This is why we normalized on an individual-patient basis. At the same time, to ensure transparency, we have kept the untransformed, per-patient measurements in the manuscript: Figure 4A shows the individual raw data points (each dot represents one patient), so readers can see the underlying values in addition to the delta-based summary.

8. In several points in the manuscript, the authors refer to gene expression changes, this is not backed up by the data and should be removed.

We have removed the unreferenced sentence in the Discussion and rephrased the remaining parts so that transcriptomic/LXR effects are clearly attributed to previously published work from our group (Zimmer et al. 2016) and are described as plausible mechanisms rather than findings of the present study.

9. The conclusion “CD has shown potential as a novel therapeutic strategy for atherosclerosis” should be reframed. Given that the current study is based on ex vivo analyses, such a therapeutic claim is not substantiated. A more appropriate phrasing would acknowledge that “CD may represent a potential candidate for further investigation as a therapeutic approach in atherosclerosis.”

Thank you for this suggestion. We have revised the conclusion exactly as proposed. This wording better reflects the ex vivo character of our study.

Minor remarks

1. Please include standard deviations and percentual values where appropriate throughout the manuscript. Additionally, indicate the sample size (n = xy) adjacent to percentage values for transparency.

Thank you for your comment. We reviewed the manuscript with this point in mind and confirm that percentages are reported together with their corresponding sample sizes where applicable, and continuous variables are presented with their dispersion measures.

2. Please rephrase the sentence “The relative number of monocytes that incorporated CC varied individually between 8 and 37%” to “The relative number of monocytes that incorporated CC ranged between 8–37%.”

We have rephrased the sentence as suggested and now state: “The relative number of monocytes that incorporated CC ranged between 8-37%.

3. Page 8: The statement “which validates the hypothesis that CD reduces CC-phagocytosis” should be relocated to the Discussion section, as it represents an interpretation rather than a result.

Thank you for pointing this out. We have removed the sentence from the Results section; the corresponding idea was already addressed in the Discussion.

4. The statement “40 patients displayed an individually significant CCΔCD while nine patients developed an increase in CC-uptake after CD-stimulation” requires revision. As the experiments were conducted ex vivo, it would be more accurate to state that “PBMCs derived from 40 patients demonstrated a significant CCΔCD, whereas cells from nine patients exhibited an increase in CC uptake following CD stimulation.”

Thank you for this suggestion. We have rephrased the sentence as suggested to better reflect the ex vivo character of our study.

5. Throughout the manuscript, please replace the term “heart catheterization” with the more precise and clinically appropriate term “coronary angiography.”

Thank you for noting this wording issue. We have revised the manuscript and replaced all occurences of “heart catherization” with the clinically more precise term “coronary angiography”.

6. The sentence “Additionally, CD increases CC solubility and promotes the formation of oxysterols, which activate liver-X-receptor (LXR)” should be either supported by a reference or removed, as there are no LXR-related experiments reported in the current study.

Thank you for drawing our attention to this. We agree that the statement requires a supporting citation. We have therefore retained the sentence but now explicitly refer to our group`s previously published work (Zimmer et al., 2016), which demonstrated that cyclodextrin solubilizes cholesterol, increases oxysterol availability and activates LXR signaling in macrophages.

---

## [Decision Letter · Decision Letter 1]

25 Nov 2025

Cyclodextrin reduces cholesterol crystal uptake by circulating monocytes in patients undergoing coronary angiography.

PONE-D-25-33939R1

Dear Dr. Luebbering,

We’re pleased to inform you that your manuscript has been judged scientifically suitable for publication and will be formally accepted for publication once it meets all outstanding technical requirements.

Kind regards,

Marc W. Merx, MD

Academic Editor

PLOS ONE

Additional Editor Comments (optional):

Reviewers' comments:

Reviewer's Responses to Questions

**Comments to the Author**

1. If the authors have adequately addressed your comments raised in a previous round of review and you feel that this manuscript is now acceptable for publication, you may indicate that here to bypass the “Comments to the Author” section, enter your conflict of interest statement in the “Confidential to Editor” section, and submit your "Accept" recommendation.

Reviewer #1: All comments have been addressed

Reviewer #2: All comments have been addressed

2. Is the manuscript technically sound, and do the data support the conclusions?

Reviewer #1: Yes

Reviewer #2: Yes

3. Has the statistical analysis been performed appropriately and rigorously?

Reviewer #1: Yes

Reviewer #2: Yes

4. Have the authors made all data underlying the findings in their manuscript fully available?

Reviewer #1: Yes

Reviewer #2: Yes

5. Is the manuscript presented in an intelligible fashion and written in standard English?

Reviewer #1: Yes

Reviewer #2: Yes

6. Review Comments to the Author

Reviewer #1: no further comments, the authors have addressed all points of concern from the original submission adequately

Reviewer #2: The authors have targeted all remarks from Reviewer 1 and me in a point-by-point rebuttal document. Most points were adequately targeted and the manuscript has improved.

7. PLOS authors have the option to publish the peer review history of their article (what does this mean?). If published, this will include your full peer review and any attached files.

Reviewer #1: No

Reviewer #2: No

---

## [Editor Report · Acceptance letter]

PONE-D-25-33939R1

PLOS One

Dear Dr. Lübbering,

I'm pleased to inform you that your manuscript has been deemed suitable for publication in PLOS One. Congratulations! Your manuscript is now being handed over to our production team.

Kind regards,

on behalf of

Prof. Dr. Marc W. Merx

Academic Editor

PLOS One